# Recent Advances on Hydrogen Evolution and Oxygen Evolution Catalysts for Direct Seawater Splitting

**Linzhou Zhuang, Shiyi Li, Jiankun Li, Keyu Wang, Zeyu Guan, Chen Liang and Zhi Xu \***

State Key Laboratory of Chemical Engineering, School of Chemical Engineering, East China University of Science and Technology, Shanghai 200237, China; lzzhuang@ecust.edu.cn (L.Z.); y11200011@mail.ecust.edu.cn (S.L.); y20200023@mail.ecust.edu.cn (J.L.); y30191172@mail.ecust.edu.cn (K.W.); y30210038@mail.ecust.edu.cn (Z.G.); y20190082@mail.ecust.edu.cn (C.L.)
**\*** Correspondence: zhixu@ecust.edu.cn

**Abstract:** Producing hydrogen via water electrolysis could be a favorable technique for energy conversion, but the freshwater shortage would inevitably limit the industrial application of the electrolyzers. Being an inexhaustible resource of water on our planet, seawater can be a promising alternative electrolyte for industrial hydrogen production. However, many challenges are hindering the actual application of seawater splitting, especially the competing reactions relating to chlorine at the anode that could severely corrode the catalysts. The execution of direct seawater electrolysis needs efficient and robust electrocatalysts that can prevent the interference of competing reactions and resist different impurities. In recent years, researchers have made great advances in developing high-efficiency electrocatalysts with improved activity and stability. This review will provide the macroscopic understanding of direct seawater splitting, the strategies for rational electrocatalyst design, and the development prospects of hydrogen production via seawater splitting. The non-precious metal-based electrocatalysts for stable seawater splitting and their catalytic mechanisms are emphasized to offer guidance for designing the efficient and robust electrocatalyst, so as to promote the production of green hydrogen via seawater splitting.

**Keywords:** seawater splitting; catalyst; oxygen evolution reaction; hydrogen evolution reaction; hypochlorite evolution reaction; anti-corrosion



## 1. Introduction

With the increasing fossil energy depletion, environmental pollution, and global warming, the development of green and renewable resources, such as wind energy, hydro energy, solar energy, and ocean energy has become a topic that countries around the world attach great importance to. Most renewable energy has the inherent problem of being intermittent, random, and volatile, thus resulting in serious abandonment of wind, light, and water. As a storable energy source, hydrogen energy has continuously promoted the transformation of traditional fossil energy into green energy by virtue of its advantages, such as flexibility, high efficiency, being carbon-free, safe, renewable, and high energy density (140 MJ/kg), which is 3 times that of oil and 4.5 times that of coal. It is regarded as a secondary energy with the greatest development potential in the future energy transformation strategy [1]. At present, the hydrogen energy industry has achieved rapid development in China and many countries around the world. In 2020, China's central government and local governments have launched more than 40 hydrogen energy industry plans and special projects, such as "Made in China 2025". The relevant key industries are included in the "14th Five-Year Plan". At the same time, the United States, Japan, South Korea, the Netherlands, Germany, France, and other EU countries have successively issued various hydrogen energy development roadmaps and hydrogen energy strategic plans [2].

Hydrogen, as the core foundation of the upstream industry chain of hydrogen energy, has many preparation methods and a wide range of sources. Hydrogen is mainly produced

by the reforming of natural gas, as well as gasification of coal and crude oil. Water splitting to produce hydrogen (and oxygen) has the advantage of producing high-purity hydrogen, but its applications are often constrained to a small scale. To date, scientists have developed various technologies to improve electrolyzers' performance and decrease their costs, including the proven alkaline electrolysis, proton exchange membrane (PEM) electrolysis, the newly developed anion exchange membrane (AEM) electrolysis, and the still immature high-temperature solid oxide electrolysis. These electrolyzers all possess cathode, which hydrogen evolution reaction (HER) processes, and anode, which oxygen evolution reaction (OER) causes. While readily available in the laboratory, a larger quantity of freshwater feeds could be an inevitable bottleneck for low-temperature water electrolyzer technologies if they would be applied widely in hot arid regions around the world, as the hot arid regions have poor access to freshwater. Unlike the scarce freshwater, seawater and the oceans represent about 96% of the total water reserves on our planet and thus are an almost unlimited resource. Therefore, water electrolyzing with eawater as direct feed could be a remarkable path for green hydrogen production [3]. Therefore, the development of high-efficiency seawater electrolysis hydrogen production technology can not only alleviate the shortage of freshwater resources but also promote the development of hydrogen energy economy and utilization of marine resources.

However, because the composition of seawater is much more complex, compared with the direct electrolysis of freshwater, seawater electrolysis faces more severe challenges. On the one hand, cations such as cobalt and magnesium in seawater are prone to generate insoluble substances during the electrolysis process on the surface of the catalyst [4], thereby reducing the reactivity of the catalyst [5,6]. On the other hand, seawater is rich in various electrochemically active anions, and these anions will directly interact with the anode during seawater electrolysis to compete with OER. Even the composition of seawater may vary greatly in different regions; the total ion concentration in seawater is always maintained at around 3.5% with pH of around 8 [7–10]. For the HER at the cathode, the lifetime of the catalyst is an unavoidable challenge in seawater electrolysis. This is due to the drastic increase in local pH near the electrode when HER occurs at the cathode. When the pH is greater than 9.5, calcium and magnesium ions in seawater will combine with hydroxide to form insoluble substances, such as calcium oxide and magnesium hydroxide. These impurities are easily attached to the surface of the catalyst, thereby hindering the direct contact between the electrolyte and the active site of the catalyst and thus severely reducing the service life of the catalyst in seawater electrolysis. Most of the reported HER catalysts would decay by more than 50% in seawater electrolysis within 24 h [11]. Besides, according to the standard redox potential of each ion in seawater, the oxidation of $Cl^-$ or $Br^-$ in seawater may compete with OER. However, since the content of $Br^-$ ions in seawater is as low as 0.00087 mol $L^{-1}$, its effect can thus be almost negligible, while the concentration of $Cl^-$ in seawater is as high as 0.5–0.6 mol $L^{-1}$ [12]. Therefore, the oxidation reaction of $Cl^-$ (chlorine evolution reaction, CER) during seawater electrolysis would easily compete with OER [13]. A comprehensive analysis of seawater electrolyzer chemistry at the anode has been presented by Dionigi et al. [14], and based on it, the chlorine evolution reaction (ClER) in an acidic solution and the hypochlorite formation (HCER) in an alkaline electrolyte could be the main competitive reactions of OER. The following two equations exhibit the chloride-related reactions in the electrolytes with low and high pH, respectively:

$$\text{ClER: } 2Cl^- \rightarrow Cl_2 + 2e^- \quad E^0 = 1.36 \text{ V vs. SHE, at pH} = 0 \tag{1}$$

$$\text{HCER: } Cl^- + 2OH^- \rightarrow ClO^- + H_2O + 2e^- \quad E^0 = 0.89 \text{ V vs. SHE, at pH} = 14 \tag{2}$$

During the CER process, the toxic chlorine gas or hydrochloric acid would be generated, which could endanger life safety and corrode reaction equipment, bringing about great production security risks. Therefore, the development of efficient catalysts for HER and OER in seawater is very important for reducing the power loss and improving the energy conversion efficiency in the hydrogen process.

In this review, we offer an overview concerning the development of different types of seawater electrocatalysts based on noble and nonprecious metals in the last few years. Moreover, we also highlight the applicable strategies to improve the activity, selectivity, and stability of the catalysts, as well as the major challenges and perspectives for the development of seawater electrolysis that could be practically applied.

## 2. HER Catalysts

Platinum (Pt) exhibits promising performance for HER in both alkaline and acidic seawater electrolyte. Many research groups have studied the HER performance of Pt in the electrolyte of the whole pH range. Other noble metals or alloys between noble metals and nonprecious metals have also attracted great interest owing to their promising binding strength of hydrogen and high stability in cruel conditions. Researchers have also developed nonprecious-metal-based catalysts to decrease the catalyst cost, reduce the energy barrier for initiating HER in tough conditions, and improve the ability of anti-fouling in seawater.

### 2.1. Precious Metal

Yang et al. strongly anchored a trace amount of precious metals on an anti-corrosion matrix (Figure 1a) [15]. For instance, the as-produced Pt/Ni-Mo needed an overpotential as low as 113 mV to obtain a high current density of 2000 mA cm$^{-2}$ in the saline–alkaline electrolyte, revealing its excellent HER performance. It showed high activity and robust durability in the strongly alkaline seawater and at temperatures up to 80 °C. Moreover, it could be mass prepared at a low cost. Mu et al. designed an ultra-low Ru-incorporated cobalt-based oxide, denoted as Ru-Co$_x$/NF [16], effectively driving the electrolysis of water at high current densities in alkaline water and seawater. The electrolyzer apparatus assembled with Ru-Co$_x$/NF only required a low voltage of 2.2 and 2.62 V to achieve 1000 mA cm$^{-2}$ in alkaline freshwater and seawater. Lee et al. proposed a catalyst of Co-CoO heterostructures incorporated with Rh atoms for efficient HER and OER catalysis in both seawater and freshwater [17]. Li et al. prepared a Pt-IrO$_2$/CC electrocatalyst modified by cyclic voltammetry, which reduced some metal cations in metal oxides (Figure 1b) [18]. This modification method resulted in the accumulation of negative charges at the active site of the metal, which significantly accelerated the HER. Notably, its precious metal load could be kept low at 36.6 µg cm$^{-2}$$_{(Ir+Pt)}$.

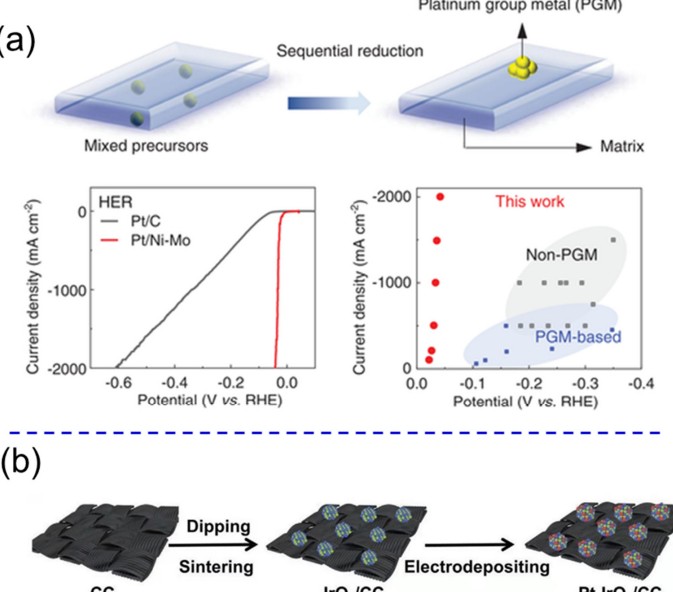

**Figure 1.** (**a**) The preparation schematic of Pt/Ni-Mo and its HER activity in alkaline seawater [15]; (**b**) illustration of the synthetic route of the Pt-IrO$_2$/CC [18].

### 2.2. Transition Metal Sulfide

Sun et al. designed the $Cu_2S@Ni$ nanorod arrays for alkaline seawater, possessing a HER current density as large as ~500 mA cm$^{-2}$ at the overpotential of <200 mV and a favorable stability [19]. The Ni-S interaction between the Ni surface and $Cu_2S$ core could optimize the hydrogen adsorption energy, so as to improve the HER activity of $Cu_2S@Ni$. Zhao et al. prepared the Co-doped $VS_2$ nanosheets through a facile one-pot solvothermal method and annealing process (Figure 2). It was demonstrated that Co replacing V in $VS_2$ nanosheets could lead to abundant S defects and rich active edge sites, and thus it exhibited the Tafel slope of 214 mV dec$^{-1}$ and stable operation duration of 12 h test in the alkaline seawater [20].

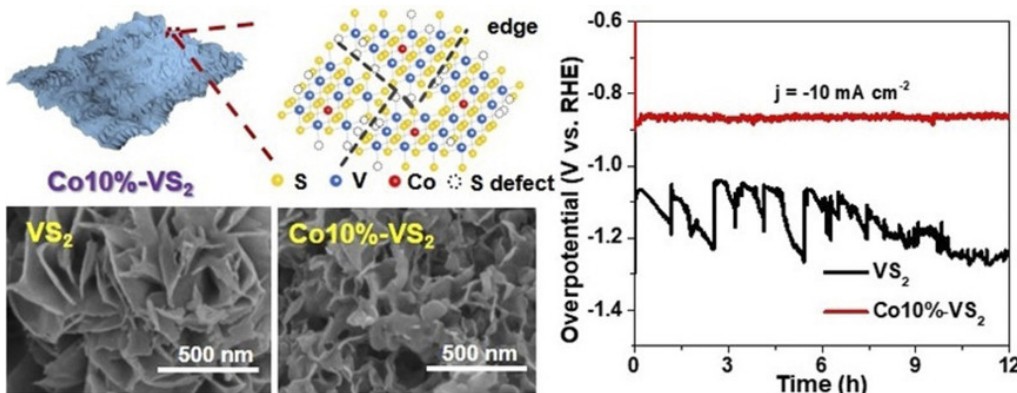

**Figure 2.** Preparation of Co-doped $VS_2$ nanosheets and its HER activity in alkaline seawater [20].

### 2.3. Transition Metal Selenide

Zhao et al. designed a cobalt selenide sample consisting of CoSe and $Co_9Se_8$ through calcining the Co foil with Se powder [21]. The Co charge state and HER activity of the catalysts could be manipulated by controlling the mass ratio of Co to Se. The authors found that a high Co charge state could favor the OER performance, while a low Co charge state could promote HER activity. Moreover, a current density of 10.3 mA cm$^{-2}$ could be achieved at 1.8 V for overall seawater electrolysis [21].

### 2.4. Transition Metal Phosphide

Xu et al. synthesized the nanoporous $C-Co_2P$ material on the basis of the Co-P-C precursor alloy by electrochemical etching (Figure 3a) [22]. The performance of $C-Co_2P$ material showed the overpotential of 30 mV at 10 mA cm$^{-2}$ in 1.0 M KOH. The authors concluded that doping C atoms could adjust the electronic structure of $Co_2P$ to form $C-H_{ad}$ intermediate that was conducive to hydrogen desorption [22]. Huang et al. prepared $Ni(OH)_2·0.75H_2O$ as the precursor by hydrothermal method and then annealed $Ni(OH)_2·0.75H_2O$ with $NaH_2PO_2H_2O$ in nitrogen atmosphere to form $Ni_5P_4$ and $Ni^{2+\delta}(OH)_{2-\delta}$, and finally, they synthesized $Ni_5P_4@Ni^{2+\delta}(OH)_{2-\delta}$ (NPNNS) (Figure 3b) [23]. $Ni_5P_4$ and $Ni^{2+\delta}(OH)_{2-\delta}$ can synergistically inhibit the properties of P-Hads bond, which impaired the HER activity, and thus, NPNNS achieved the overpotential of 87, 144, and 66 mV at 10 mA cm$^{-2}$ in alkaline freshwater, alkaline seawater, and acidic electrolyte [23]. Wang et al. prepared a catalyst $(Co-Fe_2P)$-doped transition metal phosphide in Ni foam via a hydrothermal method, which showed promising activity for both HER and OER without being corroded by chloride ions (Figure 3c) [24]. In particular, $Co-Fe_2P$ achieved overpotentials of 138 and 221 mV at 100 mA cm$^{-2}$ in 1 M KOH and simulated seawater for HER [24].

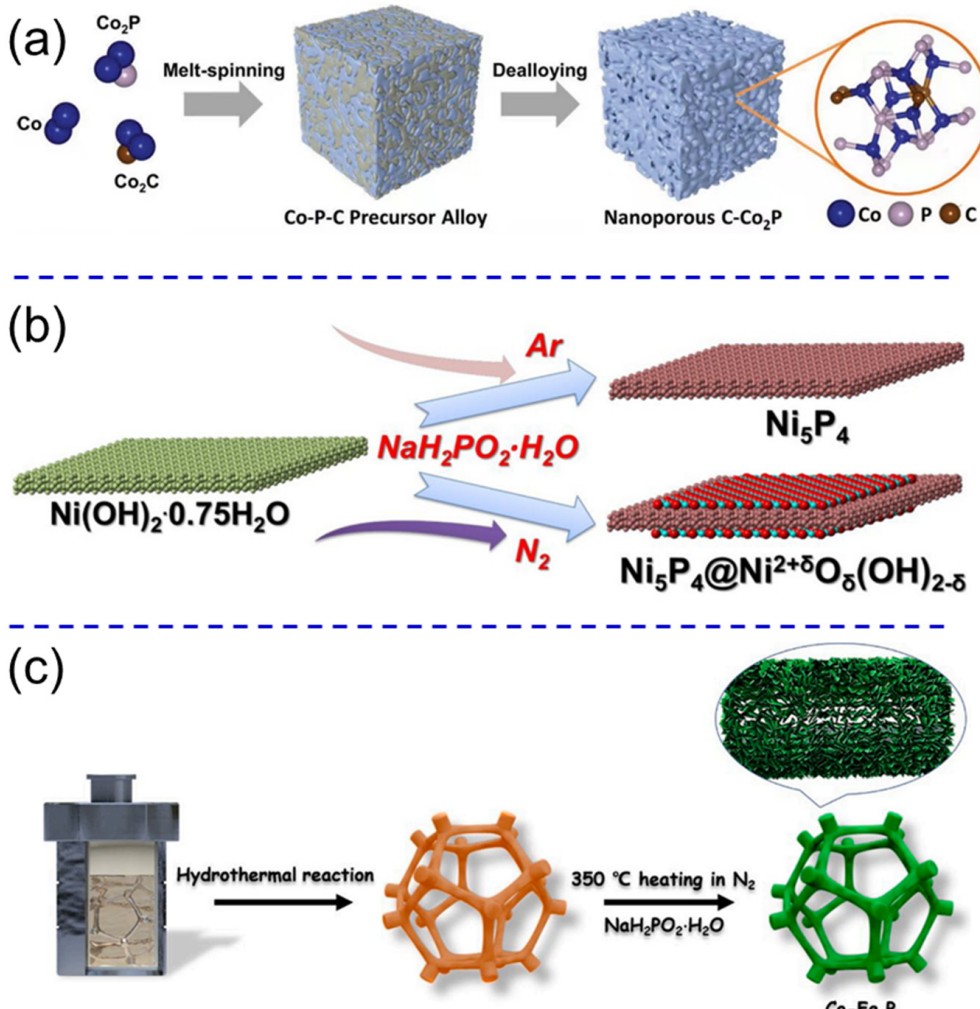

**Figure 3.** (**a**) The preparation procedure of C-Co$_2$P electrocatalysts [22]; (**b**) schematic illustration of NPNNS hybrid nanosheets [23]; (**c**) the synthesis process of the Co-Fe$_2$P electrocatalyst [24].

## 2.5. Transition Metal Carbide and Nitride

Yu et al. constructed a nanostructured NiCoN|Ni$_x$P|NiCoN catalyst HER catalysis [25], which exhibited a good HER activity and impressive stability owing to large surface area that exposed abundant active sites, high electrical conductivity, and improved intrinsic activity. It required a small overpotential of 165 mV to achieve 10 mA cm$^{-2}$ in the seawater electrolyte. Wu et al. developed a multifunctional catalytic interface to propel HER in the electrolytes of various pH and seawater (Figure 4a) [26]. The catalytic interfaces among MXene, bimetallic carbide, and hybridized carbon endowed the prepared electrocatalysts with HER activity that was comparable to the commercial Pt/C in both 1.0 M KOH and 0.5 M H$_2$SO$_4$ and even outperformed it under pH 2.2–11.2. Zang et al. demonstrated that the Ni-N$_3$ catalyst with the atomically dispersed Ni in the triple nitrogen coordination could possess efficient HER activity in alkaline electrolyte (Figure 4b) [27]. It delivered a current density as large as 200 mA cm$^{-2}$ at a lower overpotential than Pt/C, with no activity decay over 14 h. Qiu et al. reported a NiCo/Mxene-based catalyst for chlorine-free hydrogen production [28]. It required an overpotential of 49 and 235 mV to obtain 10 and 500 mA cm$^{-2}$ in 1.0 M KOH. Moreover, NiCo@C/Mxene/CF could operate stably over 120 h for HER catalysis in seawater, demonstrating its robust stability in corrosive seawater.

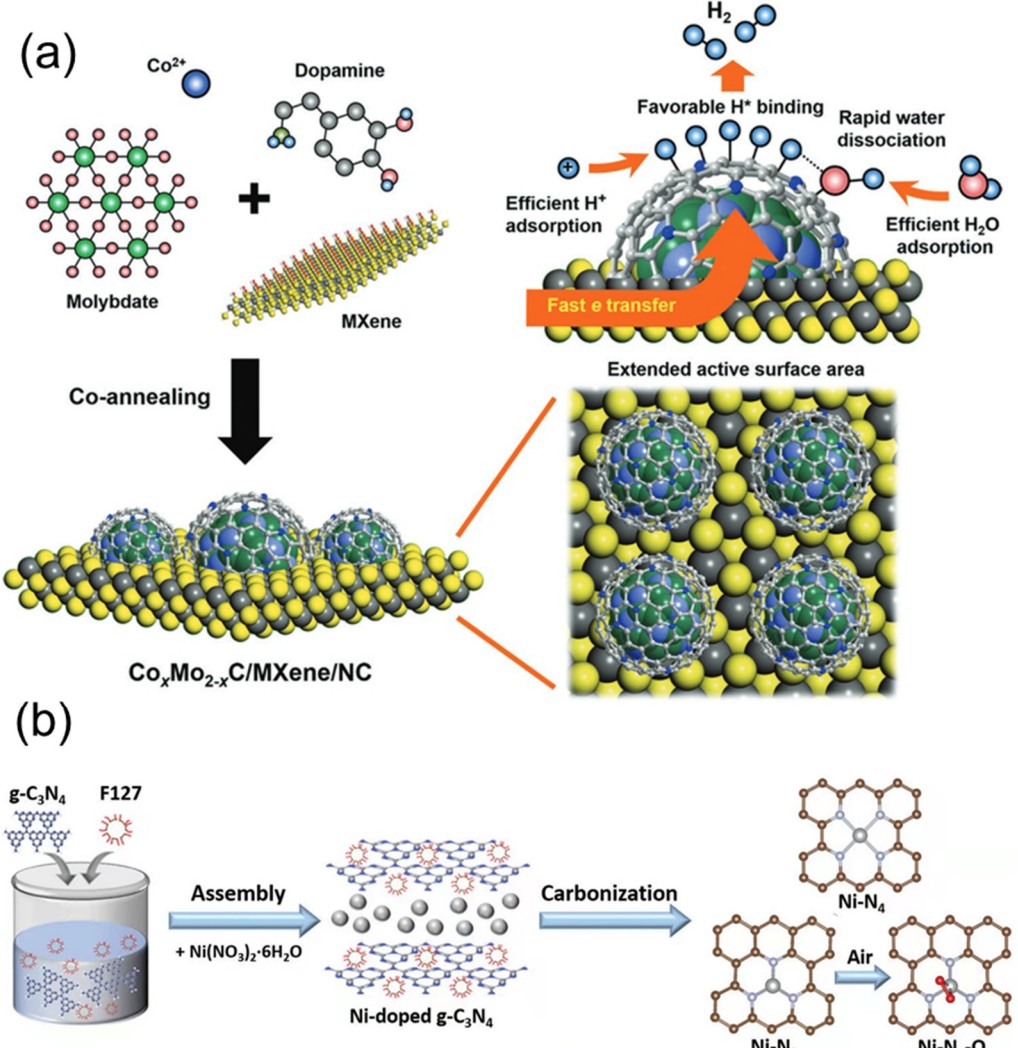

**Figure 4.** (**a**) The preparation strategy of the $Co_xMo_{2-x}C$/MXene/NC catalyst [26]; (**b**) preparation and analysis of Ni-SA/NC [27].

## 2.6. Transition Metal Alloys

Zhang et al. reported the anchoring of molybdenum-oxo functional groups on Cu substrates by in situ electrochemical reduction in CMO films [29]. The immobilized Cu plate achieved 150, 220, and 195 mA cm$^{-2}$ at an overpotential of 270 mV in 0.5 M PBS, 1 M KOH, and 0.5 M $H_2SO_4$ for HER [29]. Lu et al. established a manganese-doped nickel/nickel oxide catalyst on Ni foam through pyrolyzing Mn-based MOF [11]. In the carbonization process of Mn-MOF, Ni atoms that form the Ni foam could interact with Mn to generate a Mn-NiO-Ni heteroatom structure, leading to a strong binding between the catalyst and substrate. The Mn-doped Ni/NiO catalyst exhibited a fantastic HER activity in natural seawater, close to a commercial Pt/C catalyst, with a low overpotential of −0.17 V at 10 mA cm$^{-2}$. Durability was tested under the current density of 66.7 mA cm$^{-2}$, and it could maintain the initial potential for nearly 14 h. In addition, after cleaning the deposition on the catalyst surface, it could recover high activity [11]. Yuan et al. fabricated the NiMo film catalyst on Ni foam through the electrodepositing method. The oxidation state of Ni and Mo could be modulated by tuning the composition of aqueous solution, which can affect the catalytic activity of the NiMo film [30]. The NiO species of NiMo film is conducive to the subsequent H adsorption/desorption during the HER process, so that a small overpotential of 31.8 mV could be attained in the 1 M KOH + 0.5 M NaCl solution. Furthermore, the NiMo film also exhibited excellent durability, with negligible

increase in overpotential during the tests for 15 h [30]. Ros et al. loaded the Ni-Mo-Fe electrocatalyst on carbon to resist the corrosion of $Cl^-$ in seawater. The Ni-Mo-Fe-based catalyst reconfigured into Ni-Mo-Fe alloy and Ni:Fe(OH)$_2$ redeposits in alkaline seawater under cathodic bias [31]. Additionally, high HER activity was achieved due to the increased hydrogen adsorption strength and the suitable electronic structure of Mo surface species. In addition, the Tafel slope was 256.1, 234.2, and 201.6 mV dec$^{-1}$ in alkaline saline pure water, alkalinized seawater, and filtered seawater [31].

## 3. OER Catalysts

A large amount of $Cl^-$ exists in seawater, and the competitive CER induced by it will reduce the Faradaic efficiency of the OER. In addition, calcium and magnesium ions abundant in seawater could be easily deposited on the cathode catalyst, resulting in catalyst deactivation. In this regard, researchers have developed indirect seawater electrolysis technology, that is, first purifying seawater through a reverse osmosis membrane and then electrolyzing the obtained freshwater to produce hydrogen and oxygen. However, the reverse osmosis treatment will produce a large amount of concentrated salt brine, and its discharge will seriously harm the marine ecological environment. At the same time, the reverse osmosis membrane requires regular maintenance during the operation process, which increases the operating cost. In contrast, the direct electrolysis of seawater for hydrogen production has the advantages of low investment cost, few system engineering problems, and small device footprint, but the key is to develop high-efficiency OER catalysts with high selectivity and high stability. Numerous catalysts based on precious metals or nonprecious metals have been reported.

### 3.1. Precious Metal

Du et al. prepared IrO$_x$-Cs@BaCO$_3$ [32] whose charge transfer interaction between IrO$_x$-Cs and BaCO$_3$ could force a more negative charge on BaCO$_3$, so as to repel $Cl^-$ anions in seawater. The IrO$_x$-Cs@BaCO$_3$ with low content of Ir achieved a high mass activity of 1402 A g$^{-1}$ in the real seawater, and its stability is close to that of IrO$_2$ [32]. Haq et al. embedded Au-modified Gd-Co$_2$B nanosheets into TiO$_2$ nanosheets grown on Ti chaff (Au-GdCo$_2$B@TiO$_2$) (Figure 5a) [33]. Because of the hydrophilic bimetallic boride structure, Au-GdCo$_2$B@TiO$_2$ possessed high density of active sites and thus achieved 500 and 1000 mA cm$^{-2}$ at the overpotentials of 300 and 510 mV in alkaline seawater [33]. Ko et al. coordinated Ir atoms with heteroatoms, and even with a decreased loading of Ir to less than 6 wt% [34], the electrocatalyst could still show a low overpotential of 243 mV at 10 mA cm$^{-2}$ in 0.1 M HClO$_4$ + 3.5 wt% NaCl electrolysis. Meanwhile, in neutral synthetic seawater, the OER showed a more favorable kinetics than CER, with nearly 560 mV overpotential [34]. Gayen et al. prepared the Pb$_2$Ru$_2$O$_{7-x}$ catalyst [35] and assembled it into a membrane electrode assembly as an anode catalyst with commercial Pt/C as a cathode catalyst. The electrolyzer obtained 275 mA cm$^{-2}$ at a cell voltage of 1.80 V in the electrolyte of pH = 13 (Figure 5b). XPS confirmed that the Ru (V) oxidation state could stabilize OER intermediates through the formation of metal–oxygen bonds of large strength, while the quenching of oxygen vacancies existent in Pb$_2$Ru$_2$O$_{7-x}$ could facilitate water dissociation [35].

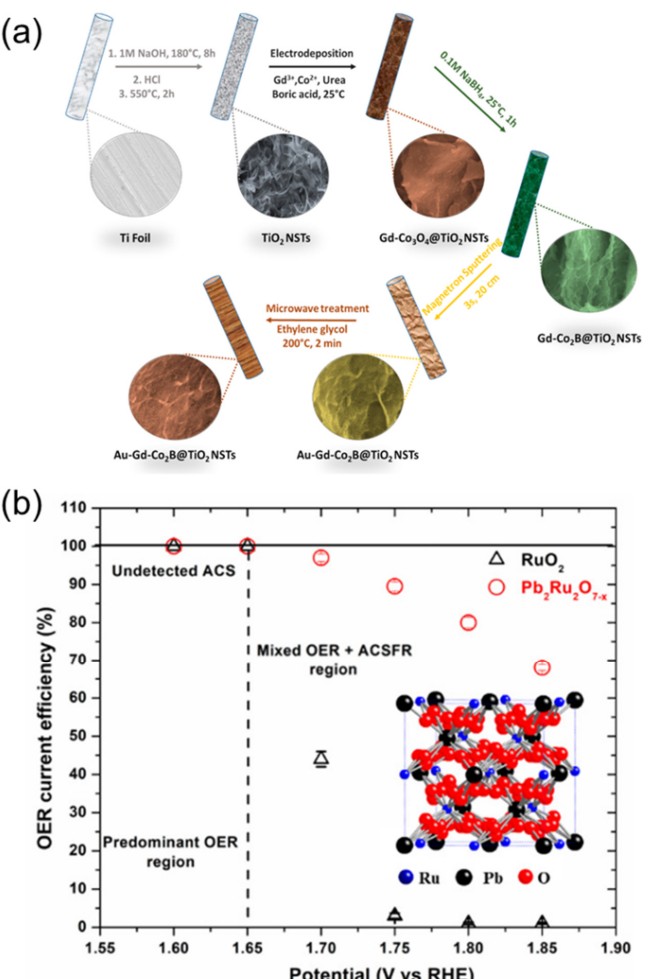

**Figure 5.** (**a**) Systematic illustration of the fabrication of Au-Gd-Co$_2$B@TiO$_2$ [33]; (**b**) the OER activity of Pb$_2$Ru$_2$O$_{7-x}$ in alkaline simulated seawater [35].

### 3.2. Transition Metal Oxide/Hydroxides

Ho et al. prepared the NiFe LDH nanosheet on a carbon cloth (CC) [24], and the NiFe-LDH/CC with a Ni/Fe ratio of 4:6 exhibited promising OER activity, reaching 10 mA cm$^{-2}$ at the overpotential of 226 mV and 238 mV in freshwater and seawater, respectively. Meanwhile, the catalyst could stably operate at 100 mA cm$^{-2}$ for over 450 h in freshwater and 165 h in seawater. Ren et al. developed an Fe$^{2+}$-driven spontaneous method of fabricating NiFe LDH [36], and the NiFe LDH could achieve 100 and 500 mA cm$^{-2}$ at the low overpotentials of 247 and 296 mV, respectively [36]. Badreldin et al. reported the preparation of S, B-(CoFeCr) and S, B-(CoFeV) oxyhydroxides for OER in alkaline and neutral-pH saline water (Figure 6a) [37]. The stronger localized Cr$^{\delta+}$(OOH)$^{\delta-}$ bonds could form an electrostatic shielding layer to impede the anionic Cl$^-$ attack. The S, B cooping could strongly enhance the hydrophilicity of catalysts, and S, B-(CoFeCr)OOH and S, B-(CoFeV)OOH could achieve 100 mA cm$^{-2}$ at the overpotentials of 353 and 408 mV, respectively [37]. Furthermore, the catalysts loaded on Ni foam could be maintained at a fixed current density of 50 mA cm$^{-2}$ in neutral pH saline environment for 50 h. Badreldin et al. also reported the fabrication of S, B-co-doped CoFe oxyhydroxide (Figure 6b) [38]. The as-prepared S,B-(CoFe)OOH-H showed a high OER selectivity of ~97% in alkaline and ~91% in neutral electrolyte at the current density of 10 mA cm$^{-2}$. Li et al. reported a simple dipping-and-heating method to synthesize FeNi oxide layer on Ni foam (FEN300) [39]. FEN300 could achieve 1000 mA cm$^{-2}$ at the overpotential of 291 mV in the electrolyte of 1.0 M KOH + seawater. Meanwhile, its current density could be maintained with no decay for 50 h [39]. DFT calculation results indicated that modifying the electronic structure via formation of the

interface between NiO and $Fe_2O_3$ could optimize the intermediate adsorption. The $e^-$-$e^-$ repulsion between $Ni^{2+}$ and oxygen intermediates adjusted and optimized the interaction of Fe with oxygen intermediates. Haq et al. prepared the $Gd-Mn_3O_4@CuO-Cu(OH)_2$ catalyst to achieve the promising OER catalysis with freestanding amorphous nanostructure [40]. The surface oxygen vacancies could modulate the electronic structure of catalytic sites and optimize the reaction intermediates' adsorption energy. Meanwhile, the hierarchical surface structure possessed high conductivity, large surface-specific area, intrinsic activity, ionic mobility, and efficient charge transfer. Its potential to deliver 500 mA $cm^{-2}$ was only 1.63 V (vs. RHE) in alkaline seawater, and its activity stability was maintained for over 75 h without any hypochlorite detected.

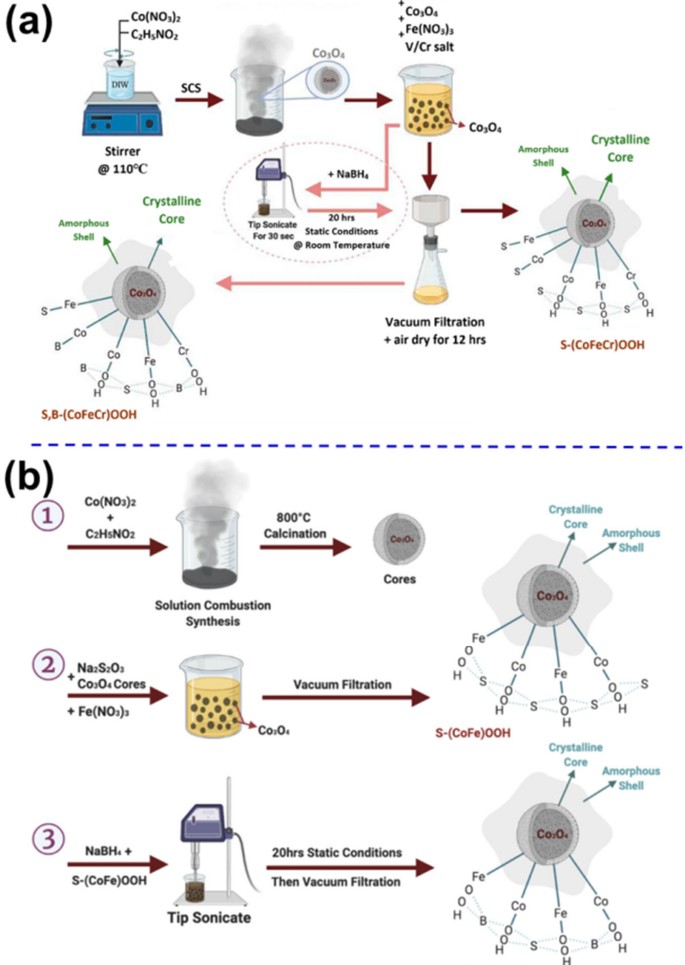

**Figure 6.** (**a**) The preparation of S, B-(CoFeCr)OOH and S, B-(CoFeV)OOH [37]; (**b**) preparation schematic of S,B-co-doped CoFe oxyhydroxide [38].

Abe et al. synthesized a manganese oxide film with rich oxygen vacancies for OER in neutral brine [41]. The catalyst was prepared by intercalating the layered manganese dioxide with $Na^+$ ions (Na | $MnO_2$) and then calcinating in the air at above 300 °C (Figure 7a). XPS results demonstrated that its oxygen vacancies began to form at 200 °C, while the valence state of Mn in the oxide clearly decreased. The Na | $MnO_x$ catalyst treated at 400 °C displayed an OER selectivity of 87% at 10 mA $cm^{-2}$. Dresp et al. prepared the NiFe-LDH electrocatalysts by a microwave-assisted solvothermal route [42], which could maintain its activity for 100 h at 200 mA $cm^{-2}$ in simulated seawater. The Faraday efficiency of the NiFe-LDH electrocatalysts was up to 88% [42]. Chen et al. took advantage of the rough surface and hydrophilic properties of wood aerogel and successfully attached the NiMoP alloys to wood aerogel (Figure 7b) [43]. NiMoP was then activated to form S,P-(Ni,Mo,Fe)OOH

nanolayers. The overpotential of S,P-(Ni,Mo,Fe)OOH/NiMoP/wood aerogel was as low as 297 mV in the alkaline seawater to achieve 500 mA cm$^{-2}$ [43]. Li et al. attached Fe(Cr)OOH and Fe$_3$O$_4$ onto Ni foam substrate to form the Fe(Cr)OOH/Fe$_3$O$_4$/NF electrocatalyst (Figure 7c). The synergistic effect among FeOOH, Fe$_3$O$_4$, and the doped Cr atoms endowed Fe(Cr)OOH/Fe$_3$O$_4$/NF with excellent OER activity, which achieved overpotentials of 198 and 241 mV at 10 and 500 mA cm$^{-2}$ in 1 M KOH. Fe(Cr)OOH/Fe$_3$O$_4$/NF could work stably at 100 mA cm$^{-2}$ for 100 h in alkaline seawater with no activity deterioration [44]. Wu et al. synthesized the CoP$_x$||CoP$_x$@FeOOH electrocatalyst with core–shell structure for OER in seawater splitting [45], which possessed a high density of active centers and a stable structural strength. CoP$_x$||CoP$_x$@FeOOH achieved the overpotentials of 480 and 637 mV at current densities of 100 and 500 mA cm$^{-2}$ in 1 M KOH + simulated seawater and showed no current attenuation for 80 h of operation without being corroded by chloride ions [45]. Jung et al. took the porous carbon scaffold as the base and uniformly attached the NiFe-LDH nanosheets to the scaffold by chemical vapor deposition (CVD) and finally synthesized NiFe-LDH-S350 after sulfurization [46]. The complex two-dimensional nanosheet structure of NiFe-LDH-S350 could achieve the overpotential of 296 mV at 100 mA cm$^{-2}$ in the electrolyte of 1 M KOH + 0.5 M NaCl for OER [46]. Gao et al. prepared the Ni/$\alpha$-Ni(OH)$_2$ heterostructure featuring the karst landform on Ni foam substrate via a facile chemical and electrochemical corrosion method. The karst morphology expanded the substrate's surface area, so as to promote the exposure of active sites, while the catalysts attached better to the substrate to possess low charge resistance and impressive durability. Therefore, the Ni/$\alpha$-Ni(OH)$_2$ catalyst showed high activity, with the overpotential of 560 mV, to achieve the OER current density of 10 mA cm$^{-2}$ in natural seawater. Moreover, its cell voltage could remain steady for 24 h owing to the protection of polyatomic sulfate layers from Cl$^-$ corrosion [47].

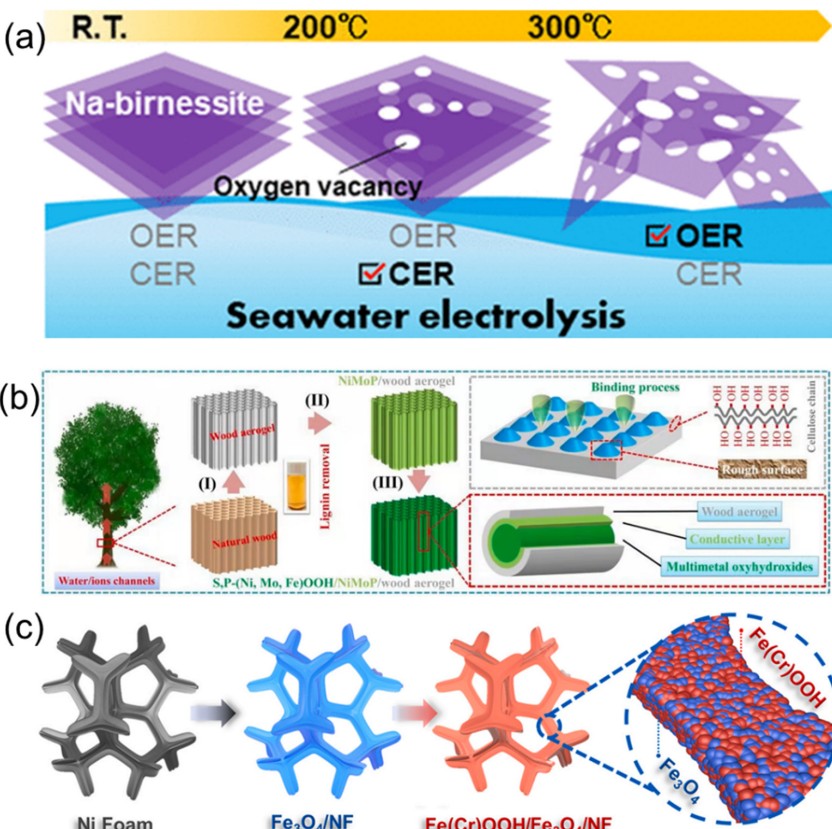

**Figure 7.** (**a**) The preparation and modulation of Na|MnO$_2$ followed by heat treatment [41]; (**b**) preparation procedure of the S,P-(Ni,Mo,Fe)OOH/NiMoP/wood [43]; (**c**) preparation of the Fe(Cr)OOH/Fe$_3$O$_4$/NF [44].

### 3.3. Transition Metal Phosphate

Song et al. used $CoCl_2$ and $NH_4H_2 \cdot PO_4$ as a metal precursor and a phosphate precursor to synthesize the $NH_4CoPO_4 \cdot H_2O$ nanosheet catalyst by chemical precipitation method (Figure 8a) [48]. In this process, Co (II) ions were activated to Co (III) species and became the active site of OER (Figure 8b). Due to its unique folding form, the two-dimensional nanosheet exhibited a large specific surface area, so as to load the high density of active centers. The overpotential of $NH_4CoPO_4 \cdot H_2O$ was 252 mV and 268 mV at 1 M KOH + seawater to achieve 10 and 100 mA cm$^{-2}$, respectively [48]. Cong et al. accomplished $Co_{0.4}Ni_{1.6}P$ nanowire arrays derived from NiCo-MOF [49] and then deposited $CeO_2$ nanoparticles on its surface to construct a novel heterostructure (Figure 8c). In this process, rich oxygen vacancies were generated to promote the catalytic activity. When testing OER activity in the alkaline simulated seawater, an overpotential as low as 345 and 394 mV was needed to achieve 10 and 100 mA cm$^{-2}$, respectively. The catalysts also showed excellent durability because of the surface $CeO_2$ nanoparticles [49].

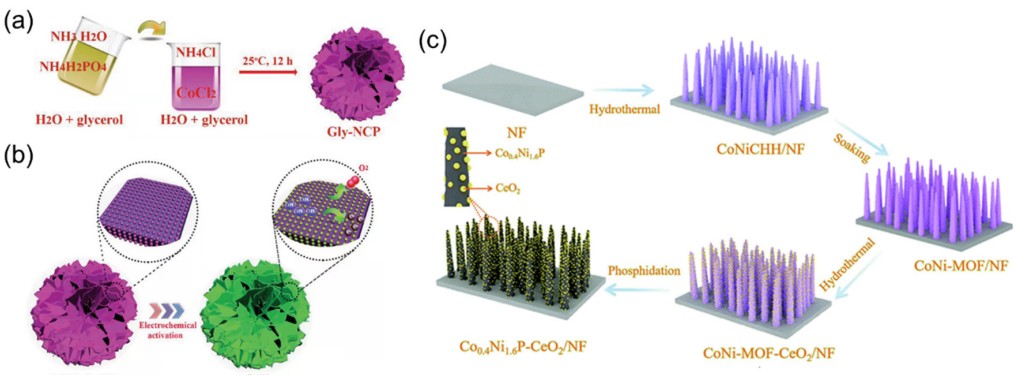

**Figure 8.** (**a**) The synthesis procedure of Gly-NCP nanosheets [48]; (**b**) the electrochemical activation process of Gly-NCP for improved OER activity [48]; (**c**) the schematic diagram for the preparation of $Co_{0.4}Ni_{1.6}P$-$CeO_2$/NF [49].

### 3.4. Transition Metal Chalcogenides

Song et al. synthesized a $MoS_2$-$(FeNi)_9S_8$ on Ni-Fe foam ($MoS_2$-$(FeNi)_9S_8$/NFF) for alkaline seawater oxidation (Figure 9a) [50]. In alkaline seawater, $MoS_2$-$(FeNi)_9S_8$/NFF could reach 100 and 500 mA cm$^{-2}$ at the overpotential of 238 and 284 mV, respectively [50]. Moreover, it could operate stably at 50 mA cm$^{-2}$ for over 70 h, demonstrating its promising stability. Its remarkable catalytic performance and stability should be attributed to the $MoS_x$ layer with plenty of defects, which could optimize the adsorption strength of H*/OH* intermediates, as well as the strong interaction between S and Ni/Fe, which could enhance the corrosion resistance ability of the substrate [50]. Chang et al. reported the Fe,P-$NiSe_2$ NFs for high-efficiency direct seawater electrolysis (Figure 9b) [51]. The doped Fe atoms were identified as the HER active sites, while the adjacent Ni atoms as OER active sites. Consequently, a large current density of 0.8 A cm$^{-2}$ could be achieved at 1.8 V, with the high OER selectivity and long-term stability for over 200 h. Hu et al. reported a $Ni_3S_2$ nanoarray-decorated Fe-$Ni(OH)_2$ for seawater oxidation [52]. The resulting Fe-$Ni(OH)_2$/$Ni_3S_2$ required a low overpotential of 269 mV to achieve 10 mA cm$^{-2}$ and showed negligible activity deterioration at 10 mA cm$^{-2}$ for 100 h. The DFT calculations indicated that the Fe sites were more favorable to OER than CER. Wang et al. uniformly grew NiCoS nanosheets onto Ni foam to form 3D self-supported catalyst (3D NiCoS NSAs) for alkaline seawater electrolysis [53]. Three-dimensional NiCoS NSAs had a high density of active sites and could effectively resist the corrosion by chloride ion. It achieved 500 and 1000 mA cm$^{-2}$ at the overpotentials of 440 and 470 mV, respectively, in alkaline seawater. Three-dimensional NiCoS NSAs can maintain the current density of 800 mA cm$^{-2}$ for 100 h in alkaline seawater with no current attenuation [53]. Yu et al. reported a facile method to synthesize S-(Ni,Fe)OOH on Ni foam within a few minutes [54]. It was demonstrated that

the S atom was incorporating in the crystal lattice, and the ultra-fast etching process led to a porous structure with huge exposure area and abundant active sites. In addition, the catalyst showed promising activity, needing the overpotential of only 378 and 462 mV to reach 1000 mA cm$^{-2}$ in 1 M KOH and 0.5 M NaCl + 1 M KOH, respectively. The S-(Ni,Fe)OOH catalyst exhibited almost no activity decay during the stability test, which could be related to the tight construction between S-(Ni,Fe)OOH and Ni foam [54].

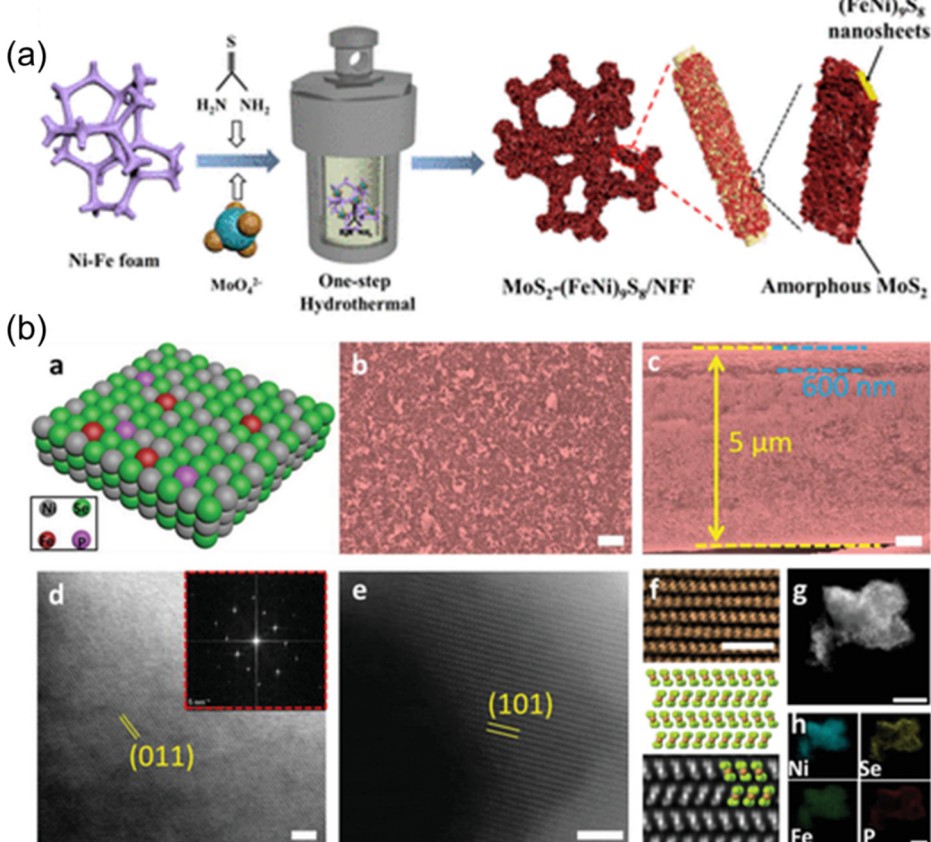

**Figure 9.** (**a**) Schematic diagram for the MoS$_2$-(FeNi)$_9$S$_8$/NFF preparation [50]; (**b**) chematic diagram for the Fe,P-NiSe$_2$ NFs preparation [51].

*3.5. Transition Metal Phosphide*

Wu et al. prepared the Ni$_2$P-Fe$_2$P nanosheet catalyst by soaking the Ni foam in an iron nitrate solution and hydrochloric acid, followed by phosphidation (Figure 10a) [55]. Owing to the abundant active sites, high intrinsic activity, and a superior transfer efficiency, this Ni$_2$P-Fe$_2$P showed a promising activity for overall water splitting, needing low voltages of 1.682 and 1.865 V to achieve 100 and 500 mA cm$^{-2}$ in 1 M KOH, respectively [55]. It could achieve 100 and 500 mA cm$^{-2}$ in 1 M KOH seawater at the voltages of 1.811 and 2.004 V, respectively [55]. Qi et al. used the Ni foam as the supporting base, wrapped the reduced graphene oxide (rGO) around it, and then attached Fe-Ni phosphides to rGO to synthesize the NiFeP/P-doped rGO/NF (NiFeP/P-rGO/NF) (Figure 10b) [56]. The overpotential of the as-obtained electrocatalyst was 290 mV and 340 mV in the electrolyte of 1 M KOH and 1 M NaCl, respectively, to achieve 100 and 400 mA cm$^{-2}$. Moreover, NiFeP/P-rGO/NF showed a high stability in seawater and could stably operate at 35 mA cm$^{-2}$ for 450 h with no significant current attenuation [56]. Yan et al. synthesized the Cu-CoP NAs/CP that attached to carbon paper by the hydrothermal method and phosphorization process (Figure 10c) [57]. Cu-doped CoP could promote the adsorption and desorption of the intermediates, thus significantly improving the catalytic performance. Thus, Cu-CoP NAs/CP needed the overpotentials of 81 and 411 mV to achieve 10 mA cm$^{-2}$ in the simulated seawater [57].

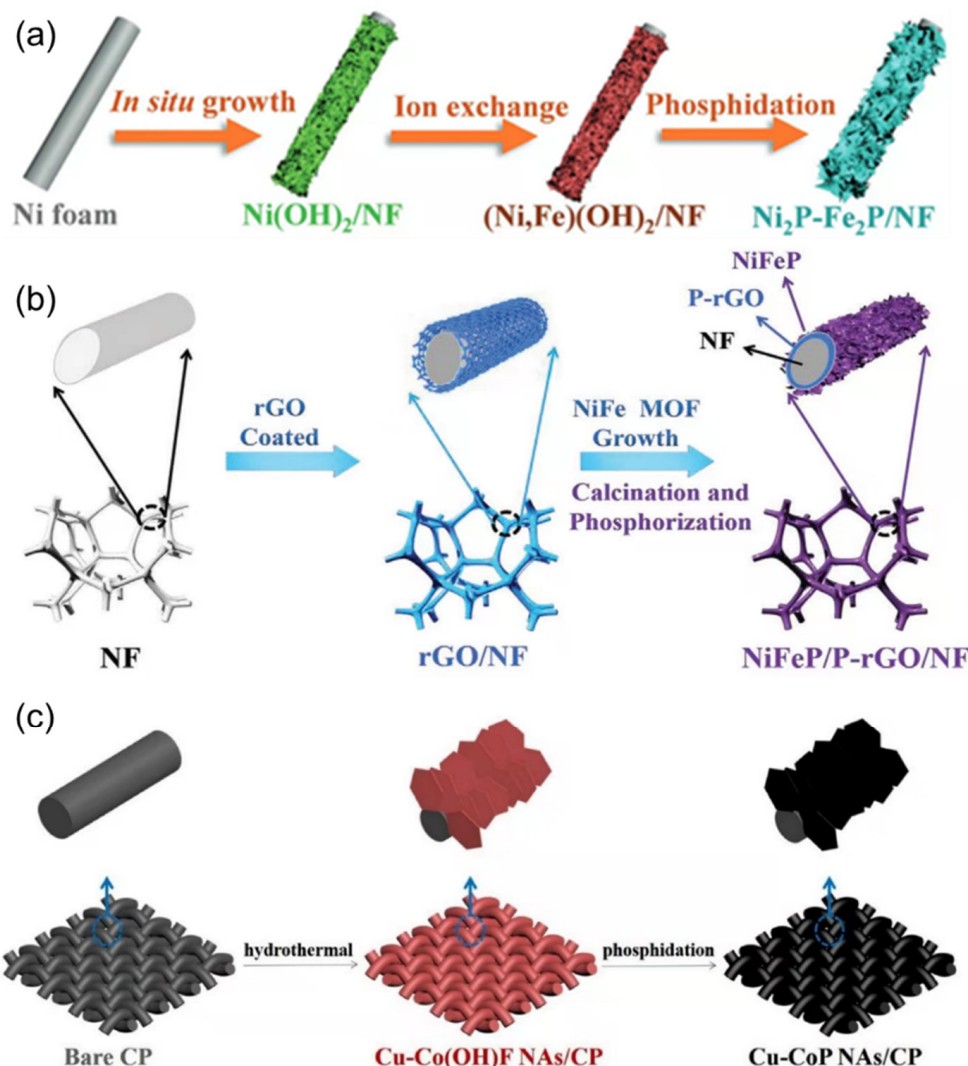

**Figure 10.** (**a**) The preparation process of $Ni_2P\text{-}Fe_2P/NF$ [55]; (**b**) the fabrication process of NiFeP/P-rGO/NF [56]; (**c**) schematic illustration of the formation of Cu-CoP NAs/CP [57].

### 3.6. Transition Metal Nitrides

Ren et al. reported a Ni-foam-based three-dimensional NiMoN@NiFeN catalyst (Figure 11) [58]. It required the overpotentials as low as 277 and 337 mV to achieve 100 and 500 mA cm$^{-2}$, respectively, and could achieve 500 and 1000 mA cm$^{-2}$ at the low voltages of 1.608 and 1.709 V when combined with NiMoN nanorods as HER catalyst at 60 °C.

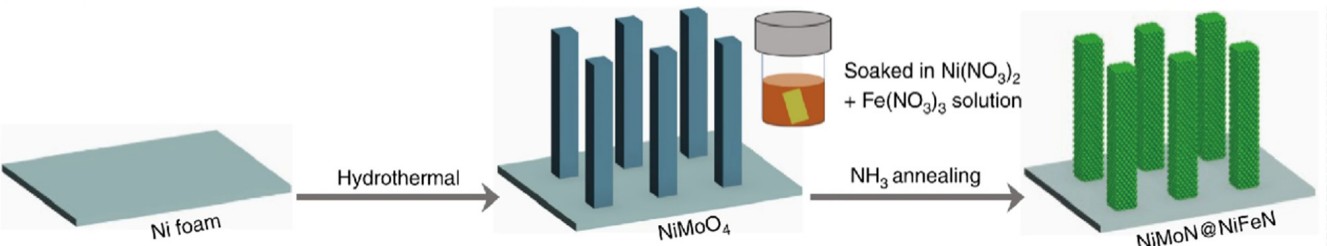

**Figure 11.** The schematic diagram of NiMoN@NiFeN preparation [58].

### 3.7. Micromolecule Additives to Form the Shielding Layer

Ma et al. reported that the service life of an anode catalyst can be significantly extended by adding $SO_4^{2-}$ ions in alkaline seawater environment (Figure 12a) [59]. The presence of

$SO_4^{2-}$ ions in the solution could preferentially adsorb on the anode surface, repel the $Cl^-$ ions by electrostatic repulsion, so as to effectively alleviate the corrosion of $Cl^-$ ions on the catalyst [59]. After adding the $SO_4^{2-}$ ions, the NiFe-LDH nanoarrays/Ni foam anode can operate stably in seawater for 500–1000 h [59]. Zhuang et al. reported a novel structural buffer engineering strategy to endow the $Co_2(OH)_3Cl$ nanoplatelets with promising long-term operation stability and an OER selectivity as high as ~99.6% in seawater splitting (Figure 12b) [60]. The lattice $Cl^-$ atoms of $Co_2(OH)_3Cl$ could play the role of structural buffers, and their continuous leaching during OER could leave vacancies for seawater $Cl^-$ invasion, so that catalyst deactivation could be avoided. Consequently, $Co_2(OH)_3Cl$ could keep 99.9% of its initial current density after 60,000 s test, while that of $Co(OH)_2$ decayed quickly by 52.7% in 7000 s [60].

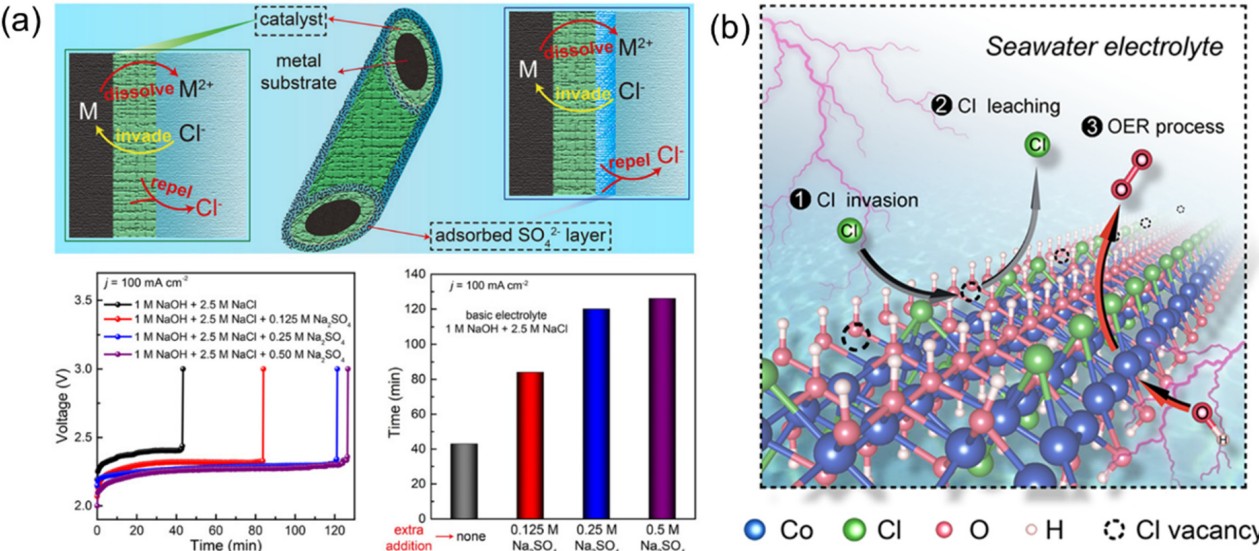

**Figure 12.** (**a**) Catalysts and electrolyte optimization to protect the metal substrate from $Cl^-$ corrosion to improve the OER stability in seawater [59]; (**b**) Mechanism of the lattice $Cl^-$ to protect the catalyst from $Cl^-$ corrosion and deactivation [17].

### 3.8. Passivation Layer Covering

Li et al. reported that the design of NiFeB/NiFeB$_x$/NiFe alloy possessed a multi-layered structure (Figure 13a) [61]. The boron species were found to exist in the form of metaborate in the outermost oxidized NiFeB$_x$ layer, and their existence could promote the generation and stabilization of the catalytic active phase γ-(Ni,Fe)OOH [61]. Meanwhile, the NiFeB$_x$ interlayer could effectively prevent the anode material from excessive oxidative corrosion in the electrolyte containing chloride ions. Chen et al. discovered a PPy and tannic acid (TA) modified hollow MIL-88(FeCoNi) catalyst (Figure 13b) [62]. TA played an important role in etching MIL-88 to form a hollow structure, which exposed abundant active sites, while the released $Fe^{3+}$ promoted the polymerization of pyrrole to enhance conductivity and stability. Owing to the coordination of PPy and TA, the HMIL-88@PPy-TA catalyst presented an extraordinary activity with a low overpotential of 370 mV at 1000 mA cm$^{-2}$ in simulated seawater. In addition, under a constant current density of 100 mA cm$^{-2}$, it could maintain an initial overpotential for 100 h. It was proved that PPy protected the HMIL-88 from $Cl^-$ corrosion, and the reconstruction of MIL-88(FeCoNi) to metal hydroxide during OER was beneficial for the maintenance of activity [62].

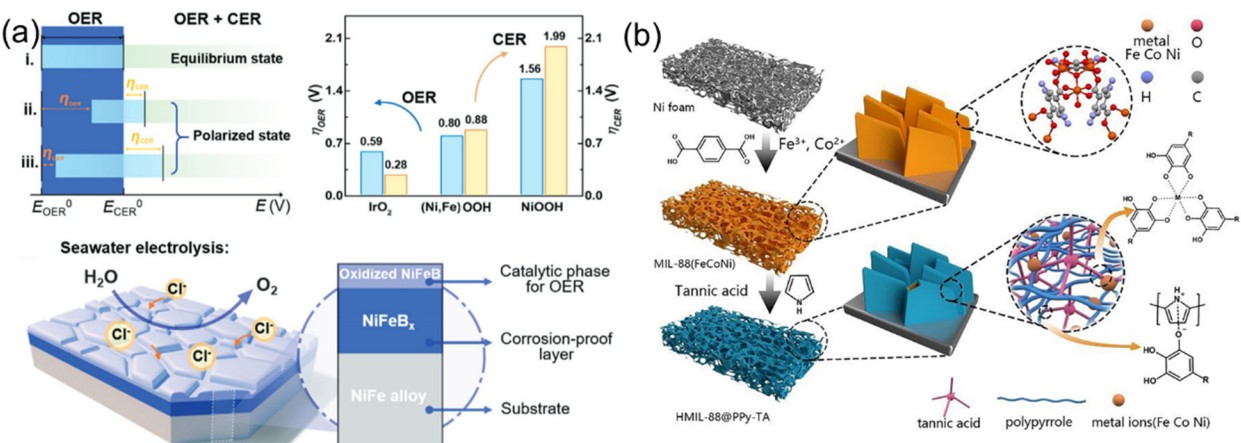

**Figure 13.** (**a**) Schematic diagram of the multilayer oxidized NiFeB/NiFeBx/NiFe alloy [61]; (**b**) Preparation schematic of HMIL-88@PPy-TA catalyst [62].

## 4. Conclusions and Perspective

Nowadays, freshwater is becoming increasingly rarer, while the earth's oceans could provide an unlimited seawater resource. Against this global background, direct seawater electrolysis has become a promising electricity/hydrogen conversion and storage technology. It is expected to be effective in the regions with abundant renewable electricity and sufficient access to ocean seawater. However, because of the undesired electrochemical processes associated with the contaminants during seawater splitting, primarily the chloride chemistry, many research groups made intensive efforts in the exploration of robust and selective electrodes. Although researchers have had great achievements in seawater electrolysis in recent decades, various challenges should be overcome to fully promote seawater electrolysis for industrial $H_2$ production. The first challenge is to synthesize the highly active electrocatalysts, which could suppress $Cl^-$ ions-associated electrochemical processes. They should exhibit high stability against corrosion by $Cl_2$, $ClO^-$, and so on. These strategies include doping various elements to modulate the electronic structure of active sites, tuning the binding strength of intermediates, synthesizing chlorine-blocking layers to promote OER, and constructing unique morphology. Second, the abundant ions in seawater could hinder the electrocatalytic reactions and result in unsatisfactory electrocatalytic performance. The promising electrode materials should effectively use the ions in seawater to improve the seawater electrolysis performance. Third, accurate identification and characterization of active sites should be brought to the forefront to understand the underlying mechanism and design the targeted electrocatalysts. Fourth, DFT calculations may play an important role in uncovering the reaction mechanisms and active sites of catalysts. The volcano plots, d-band center theory, and adsorption-free energy proposed by the computational simulations could benefit the study of intrinsic electrocatalytic properties. Meanwhile, the molecular dynamic simulation and finite element modeling may be applied to the effect of pore structure on the mass transfer and electrocatalytic activity of the catalysts. Lastly, the emerging technologies, such as machine learning, high-throughput theoretical computing, and artificial intelligence, may also be used to obtain the new high-level descriptors, so as to boost the development of catalyst design strategies. The construction of a clear structure–activity relationship could guide the rational design and fabrication of high-efficiency electrocatalysts for direct seawater splitting, so as to promote hydrogen production by seawater electrolysis.

**Author Contributions:** The manuscript was written through contributions of all authors. L.Z. and Z.X.: Conceptualization, Investigation, and Supervision. L.Z. and S.L.: Writing—original draft and image processing. J.L., K.W., Z.G. and C.L.: Validation, Resources, Investigation, Writing—review and editing. L.Z., S.L. and Z.X.: Visualization, Writing—review and editing. All authors have read and agreed to the published version of the manuscript.

**Funding:** This work was financially supported by the National Natural Science Foundations of China (Grant No. 21908054 and 22005098), and Central Government Funds for Guiding Local Science and Technology Development (Grant No. 2021Szvup040).

**Institutional Review Board Statement:** Not applicable.

**Informed Consent Statement:** Not applicable.

**Data Availability Statement:** Not applicable.

**Conflicts of Interest:** The authors declare no conflict of interest.

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
