# Peer review of "Recent Advances on Hydrogen Evolution and Oxygen Evolution Catalysts for Direct Seawater Splitting"

_coatings, doi:10.3390/coatings12050659_

Round 1

Reviewer 1 Report

In this manuscript, the authors reviewed recent articles on new HER and OER catalysts for direct seawater splitting. The writing is of acceptable quality, and the importance of the topic is appropriately addressed in the introduction. The catalysts were classified according to their composition (noble metal, non-noble metal, etc.) and discussed accordingly. Overall, however, I feel it is difficult to see novel and deep comments in this manuscript. The discussion is too superficial and mostly limited to summarizing the literature. The authors should provide more insights into the specific mechanisms underlying the high performance for direct seawater splitting, and also elaborate on their suggestions for future research directions in the field.

Author Response

To Reviewer 1

  1. In this manuscript, the authors reviewed recent articles on new HER and OER catalysts for direct seawater splitting. The writing is of acceptable quality, and the importance of the topic is appropriately addressed in the introduction. The catalysts were classified according to their composition (noble metal, non-noble metal, etc.) and discussed accordingly. Overall, however, I feel it is difficult to see novel and deep comments in this manuscript. The discussion is too superficial and mostly limited to summarizing the literature. The authors should provide more insights into the specific mechanisms underlying the high performance for direct seawater splitting, and also elaborate on their suggestions for future research directions in the field.

Reply: Thanks for your insightful suggestion.

For revealing the mechanism of OER, in Page 3, the following discussion has been added.

“Based on the former theoretical considerations of Bennett and Trasatti, Dionigi et al. presented a thorough analysis of the anodic seawater electrolyzer chemistry in 2016 [14]. On the basis of this, the chlorine evolution reaction (ClER) at low pH and the hypochlorite formation (HCER) in high pH solution are the main OER competing reactions. Equations 1 and 2 exhibit the corresponding chloride chemistry reactions at low and at high pH, respectively:

ClER: 2Cl- → Cl2+ 2e-  E0=1.36 V vs SHE, pH=0    (1)

HCER: Cl- + 2OH-→ClO- + H2O + 2e-  E0=0.89 V vs SHE, pH=14       (2)”

In Page 3, the following discussion about HER catalysts has been added.

“Platinum (Pt) exhibits the best performance for HER in both acid and alkaline sea-water electrolyte, exhibiting the lowest overpotential. Using the computational simula-tion, a number of research groups understand the behavior of HER on platinum in the entire pH range. Other noble metals or alloys between noble metals and nonprecious metals also have received extensive attention due to their Pt-like hydrogen binding strength and promising stability in harsh conditions than Pt. While studying no-ble-metal-based catalysts, researchers have also modified noble-metal-free catalysts to reduce the energy barrier for triggering HER in harsh conditions and improve the ability for anti-fouling.”

In Page 8, the following discussion about OER catalysts has been added.

“A large amount of Cl- exists in seawater, and the competitive CER induced by it will reduce the Faradaic efficiency of the OER. In addition, calcium and magnesium ions abundant in seawater could be easily deposited on the cathode catalyst, resulting in catalyst deactivation. In this regard, researchers have developed indirect seawater electrolysis technology, that is, first purifying seawater through a reverse osmosis membrane, and then electrolyzing the obtained fresh water to produce hydrogen and oxygen. However, the reverse osmosis treatment will produce a large amount of concentrated salt brine, and its discharge will seriously harm the marine ecological environment. At the same time, the reverse osmosis membrane requires regular maintenance during the operation process, which increases the operating cost. In contrast, the direct electrolysis of seawater for hydrogen production has the advantages of low investment cost, few system engineering problems, and small device footprint, but the key is to develop high-efficiency OER catalysts with high selectivity and high stability.”

With respect to the suggestions for future research direction, the following discussion has been added.

“First, the key challenge is to design the highly active electrocatalysts that are capable of suppressing Cl- ions associated electrochemical processes. The electrocatalysts for seawater electrolysis should possess high stability against corrosion by Cl2, ClO- and so on. These strategies include doping different elements to tuning the electronic structure of active sites, modulating the adsorption or desorption binding strength of intermediates, synthesizing layers blocking, chlorine to achieve the selective reaction of OER and constructing unique morphology. Second, the abundant ions in seawater could hinder the electrocatalytic reactions in seawater, and result in undesirable electrocatalytic performance. The promising electrode materials should effectively use the ions in the seawater to produce highly valuable products and simultaneously improve the electrocatalytic seawater splitting performance. Third, accurate identification and characterization of catalytic active sites should be given priority to understand the catalytic mechanism at the atomic level and design the targeted electrocatalysts. HAADF-STEM and XAFs techniques should be indispensable tools for understanding the configuration and charge transfer process of active center, analyzing coordination model and reaction mechanism. Fourth, DFT calculations show great potential in revealing the reaction mechanisms and active sites of catalysts. The major theoretical explanation of intrinsic electrocatalytic properties involves the volcano plots, d-band center theory, and adsorption free energy proposed from the computational simulations. The molecular dynamic simulation and finite element modeling may be applied to the effect of pore structure on the mass transfer and electrocatalytic activity of the catalysts. Last, the emerging technologies, such as machine learning, artificial intelligence, and high-throughput theoretical computing may be also used to discover new high-level descriptors, enabling the development of catalyst design strategies. The construction of a clear structure-activity relationship could direct the rational design and fabrication of highly efficient seawater electrocatalysts, so as to promote the hydrogen production by seawater electrolysis.”

The relevant information has also been marked in blue.

Reviewer 2 Report

In the present manuscript, authors present a review reporting the latest advances on hydrogen evolution and oxygen evolution catalysts for direct seawater splitting. The work is well organised, focusing on different classes of materials for each application. Authors mainly report the overpotential and durability performance for most of the catalysts. I acknowledge that authors present the latest advances, as most of the references are from the last 3 years, thus they capture the recent trends. I believe that this review will be of interest for the readers of the present journal and recommend its acceptance for publication.

I only have the following minor suggestion regarding the outlook. Authors point towards 3 key features that catalysts for sea water splitting should fulfill. Thus, they should give specific examples of candidate materials that most efficiently acomplish these requirements.

Author Response

To Reviewer 2

In the present manuscript, authors present a review reporting the latest advances on hydrogen evolution and oxygen evolution catalysts for direct seawater splitting. The work is well organised, focusing on different classes of materials for each application. Authors mainly report the overpotential and durability performance for most of the catalysts. I acknowledge that authors present the latest advances, as most of the references are from the last 3 years, thus they capture the recent trends. I believe that this review will be of interest for the readers of the present journal and recommend its acceptance for publication. I only have the following minor suggestion regarding the outlook. Authors point towards 3 key features that catalysts for seawater splitting should fulfill. Thus, they should give specific examples of candidate materials that most efficiently accomplish these requirements.

Reply: Thanks a lot for your supportive comments.

Indeed, we have reviewed a series of HER and OER catalysts based on noble metals or nonprecious metals for seawater splitting that reported in the last three years, which exhibited promising electrocatalytic activities. However, most of them can hardly meet the requirement of the three features simultaneously. Therefore, in the Conclusion and Perspective section, we summarize the strategies, include doping different elements to tuning the electronic structure of active sites, modulating the adsorption or desorption binding strength of intermediates, synthesizing layers blocking, chlorine to achieve the selective reaction of OER and constructing unique morphology, to further improve the activities of the reported catalysts. The DFT, MD, and FEM calculation tools are also suggested to be applied in the development of the promising seawater splitting catalysts.

The relevant discussion has been added in the manuscript and marked in blue.

Reviewer 3 Report

The present review article provides insights into the understanding of the non-precious metal-based electrocatalyst design for direct seawater splitting and the future perspectives of hydrogen production. The topic is very interesting and will attract many researchers working in the similar field. The manuscript can be accepted if the authors address the following minor issues. 
1.    Introduction part should emphasis on the various present approaches for the production of hydrogen and compare their merit and demerits with seawater electrolysis.
2.    Author should include the mechanism of seawater splitting with proper schematic figure, which could help readers to better understand and design the electrocatalysts for OER and HER.
3.    Author have summarized several non-precious catalysts but at the beginning of each subsection the importance, significance and selection of particular electrocatalysts for HER or OER should be mentioned. 
4.    Figure quality should be improved. For an example Fig. 5, 6, 8 and so on.
5.    The conclusion and perspective section is too general and lacks the clear future directions. 

Round 2

Reviewer 1 Report

The authors have appropriately revised their manuscript in response to my concerns, and I support publication at this stage.